# Whole-Genome Sequence and Characterization of *Ralstonia solanacearum* MLY102 Isolated from Infected Tobacco Stalks

**DOI:** 10.3390/genes15111473

**Published:** 2024-11-15

**Authors:** Guan Lin, Juntao Gao, Junxian Zou, Denghui Li, Yu Cui, Yong Liu, Lingxue Kong, Shiwang Liu

**Affiliations:** 1School of Biological & Chemical Engineering, Zhejiang University of Science & Technology, Hangzhou 310023, China; 212103817039@zust.edu.cn (G.L.); 212103817003@zust.edu.cn (J.G.); 222203855049@zust.edu.cn (J.Z.); 222303855018@zust.edu.cn (D.L.); 222403857045@zust.edu.cn (Y.C.); liushiwang@zust.edu.cn (S.L.); 2Institute for Frontier Materials, Deakin University, Geelong, VIC 3216, Australia

**Keywords:** *Ralstonia solanacearum*, isolation, genome, comparison

## Abstract

Background/Objectives: Bacterial wilt disease is a soil-borne disease caused by *Ralstonia solanacearum* that causes huge losses to crop economies worldwide. Methods: In this work, strain MLY102 was isolated and further identified as *R. solanacearum* from a diseased tobacco stalk. The genomic properties of MLY102 were explored by performing biochemical characterization, genome sequencing, compositional analysis, functional annotation and comparative genomic analysis. Results: MLY102 had a pinkish-red color in the center of the colony surrounded by a milky-white liquid with fluidity on TTC medium. The biochemical results revealed that MLY102 can utilize carbon sources, including D-glucose (dGLU), cane sugar (SAC) and D-trehalose dihydrate (dTRE). Genome sequencing through the DNBSEQ and PacBio platforms revealed a genome size of 5.72 Mb with a G+C content of 67.59%. The genome consists of a circular chromosome and a circular giant plasmid with 5283 protein-coding genes. A comparison of the genomes revealed that MLY102 is closely related to GMI1000 and CMR15 but has 498 special genes and 13 homologous genes in the species-specific gene family, indicating a high degree of genomic uniqueness. Conclusions: The unique characteristics and genomic data of MLY102 can provide important reference values for the prevention and control of bacterial wilt disease.

## 1. Introduction

*Ralstonia solanacearum* is a Gram-negative bacterium belonging to the phylum *Proteobacteria* [1]. The bacterium is rod-shaped, rounded at both ends and bears 1–4 polar flagella with a cell length of approximately 0.5–1.5 μm [2]. *R. solanacearum* has complex physiological and biochemical properties and can infect more than 450 species of plants in more than 54 families, including common crops such as chili, tomato, tobacco, potato, peanut, ginger, mulberry and eucalyptus [3,4,5]. Plants infected with *R. solanacearum* develop bacterial wilt disease and gradually wilt and die. Bacterial wilt disease is widely distributed in tropical, subtropical and temperate regions and causes severe economic damage to crops [6]. Tobacco is a widely grown cash crop. Bacterial wilt disease was first identified in tobacco in 1880 and is categorized as an important disease in tobacco production because of its potential threat to tobacco [7]. The infestation of tobacco with *R. solanacearum* causes significant economic losses and yield reductions, contributing to 10–30% of production losses globally [8]. Therefore, studying the *R. solanacearum* genome helps us understand its pathogenicity and the development of resistance.

*R. solanacearum* is spread through soil, it is widely distributed and very viable and it can survive for several years without any nutrients [9]. *R. solanacearum* can be classified into four phylotypes based on geographic location: phylotype I (Asia), phylotype II (America), phylotype III (Africa) and phylotype IV (Indonesia) [10]; another type of classification is based on the use of carbohydrates (lactose, maltose, fiber diatoms and mannitol, etc.), and *R. solanacearum* can be classified into five biochemical variants [1]. In addition, after infesting a plant, *R. solanacearum* can return to the soil from fallen tissues, causing a secondary infestation [11].

The whole-genome sequencing of *R. solanacearum* can provide a basis for the study of its pathogenesis at the genome level. GMI1000, a model strain, was isolated from tomato plants, and its complete nucleotide genome sequencing has been completed [12]. These findings provide important reference information for subsequent studies on the evolution of the diversity of *R. solanacearum* and the pathogenesis of related diseases. The genome of *R. solanacearum* is approximately 5.8 Mb in size and consists of two circular replicons (a circular chromosome and a circular megaplasmid), and a few have a small plasmid (e.g., Rs-P.362200) [2]. The pathogenicity of bacterial wilt disease is closely related to its virulence factors. Virulence factors include extracellular polysaccharides, effector proteins and flagella [13]. By invading plant conduits, bacterial wilt disease synthesizes large amounts of extracellular polysaccharides, which block vascular tissues to impede water transport and ultimately lead to plant wilting and death. The more studied virulence factor is the type III secretion system effector protein (T3SS) [14]. *R. solanacearum* injects the T3SS into plant cells via its syringe-like Type III secretion system, and its effects include interfering with the basic defense system of the plant, interfering with plant metabolic processes, promoting infestation, and stimulating immune responses of the host plant [2,15,16,17]. Currently, the T3SS genes of *R. solanacearum*, commonly known as Rip genes, include 102 Rip genes and 16 T3SS candidate genes [18]. To date, the NCBI database has published the complete draft genomes of 507 *R. solanacearum* strains, which mainly include isolates from tomato, tobacco, chili and sesame [4]. The use of these *R. solanacearum* genomes can help us better understand the regulatory mechanisms of *R. solanacearum* and provide important reference values for controlling bacterial wilt disease at the gene level.

There are large differences in virulence among different *R. solanacearum* strains. In this study, we performed physiological, biochemical and whole-genome sequencing analyses (Process: Appendix A) of the *R. solanacearum* strain MLY102, which was isolated from an onset tobacco sample from a plot with a high incidence of bacterial wilt disease. Due to the economic losses caused by *R. solanacearum*, we hope to find some basis for the high pathogenicity of this strain through functional annotation and comparative genomics, which will provide key data for the development of new control strategies. This is not only crucial for the sustainable development of the tobacco industry but also has broad implications for disease prevention and control in other crops.

## 2. Materials and Methods

### 2.1. Materials and Equipment

TTC medium, PDA medium and yeast extract were obtained from Hangzhou Microbial Reagent Co., Ltd. (Hangzhou, China). The universal primers Eubac27F (5′-AGAGTTTGATCCTGGCTCAG-3′) and Eubac1492R (5′-GGTTACCTTGTTACGACTT-3′) were obtained from Beijing Tsingke Biotechnology Co., Ltd. (Beijing, China). A scanning electron microscope (SEM) SU1510 was obtained from Hitachi (Mitoshi, Japan), and NanoSuit^®^ solution I was obtained from Nisshin EM Co., Ltd. (Tokyo, Japan). The automated bacterial identification instrument VITEK 2 Compact and GN Identification Card were obtained from BioMérieux Corporate (Lyon, France). The single-molecule real-time (SMRT) PacBio RS II sequencing instrument was obtained from Pacific Biosciences (Menlo Park, CA, USA).

### 2.2. Isolation and Identification of R. solanacearum MLY102

#### 2.2.1. Isolation of *R. solanacearum* MLY102

Samples of diseased tobacco stalks were collected from farmland in Tianzhu County, Guizhou Province, China, which has a high incidence of bacterial wilt disease throughout the year. The stem of a diseased tobacco plant was taken, its surface sterilized, and the diseased tissue within the stem removed and suspended in sterile water. A bacterial suspension was obtained via sufficient shaking; it was then appropriately diluted and coated on NA medium and incubated at 30 °C for 48 h. The stem of the diseased tobacco plant was then incubated at 30 °C for 48 h. The diseased tobacco stem was then disinfected with a surface disinfectant. The 16S ribosomal RNA gene was amplified with the universal primers Eubac27F (5′-AGAGTTTGATCCTGGCTCAG-3′) and Eubac1492R (5′-GGTTACCTTGTTACGACTT-3′). The DNA sequences were sequenced and blasted in GenBank to identify the species affiliation of the isolate MLY102.

#### 2.2.2. Biochemical Characterization Tests of *R. solanacearum* MLY102

A fresh pure culture of *R. solanacearum* MLY102 was collected, suspended in saline solution (0.45–0.50% NaCl), and transferred to a VITEK^®^ 2 suspension tube containing 3 mL of saline, with the McFarland turbidity adjusted to 0.50–0.63. The suspension tube was then attached to the GN card, and the biochemical reaction pattern was analyzed using the VITEK^®^ 2 Compact System.

### 2.3. R. solanacearum MLY102 Genome Sequencing and Assembly

The genome of strain MLY102 was sequenced at the Beijing Genome Institute (BGI, Shenzhen, China) via the PacBio and DNBSEQ platforms. DNA from MLY102 was extracted using the MGIEasy Microbiome DNA Extraction Kit. The extracted DNA was fragmented to the target fragment range (15–20 K) using a Megaruptor on the PacBio platform and then precisely recovered via Sage ELF to obtain libraries with high-fidelity reads for sequencing. The PacBio Sequel platform uses SMRT cells to generate subreads. Subreads shorter than 1000 bp in length were filtered out and then self-corrected via Canu v1.5 [19]. To obtain high-confidence assembled sequences, single-base corrections were performed via GATK v1.6-13 [20].

### 2.4. Genome Component Prediction

Gene prediction of *R. solanacearum* was performed via Glimmer v3.02 [21]. RNAmmer v1.2, tRNAscan v1.3.1 and Rfam v9.1 were used to recognize rRNA, tRNA and sRNA, respectively [22,23,24]. Tandem repeat sequences were obtained via Tandem Repeat Finder v4.04 and screened for microsatellite and minisatellite sequences therein on the basis of the length and number of repeat units [25].

### 2.5. Functional Annotation of the Genome

The genes were blasted with Diamond v0.8.31 in different databases [26], such as VFDB (virulence factor database) v2023-4-28 [27], ARDB (Antibiotic Resistance Genes Database) v1.1 [28], CAZy (Carbohydrate-Active enZYmes Database) v2022-9-15 [29], Swiss-Prot vrelease-2022-03 [30], COG (Cluster of Orthologous Groups of proteins) v2020-11-25 [31], CARD (The Comprehensive Antibiotic Resistance Database) v3.0.9 [32], GO (Gene Ontology) v2019-07-01 [33], KEGG (Kyoto Encyclopedia of Genes and Genomes) v106.0 [34], T3SS (Type III secretion system Effector protein) v1.0 [35] and NR (Non-Redundant Protein Database databases, v2023-4-20). The highest quality comparison results were selected for gene annotation.

### 2.6. Comparative Genomic Analysis

#### 2.6.1. Structural Variation (Synteny)

MLY102 and 10 other reference strains (CFBP2957, CMR15, GMI1000, PSI07, Po82, RS-CIAT078, RS-Rs5, RS-SL2064, RS-T95 and RS-UW251) were analyzed. The gene sequence comparison was performed using MUMmer v3.22 [36]. Protein comparison was performed using a BLASTp comparison of the target bacterial protein set to the reference bacterial protein set. Finally, according to its location information, it was reduced to the same scale and labeled on the figure.

#### 2.6.2. Genome-Wide Similarity Analysis

Genome-wide similarity was analyzed on the basis of average nucleotide identity (ANI), and fastANI v1.32 software was used to perform the analysis (parameters: -c 1024) [37].

#### 2.6.3. Core and Pan Genome Analysis

Analysis of the core and pan genomes was performed via CD-HIT v4.6.6 (parameters: -c 0.5 -n 3 -p 1 -g 1 -d 0) [38]. The protein-encoding gene sets of all the strains to be analyzed were subjected to CD-HIT clustering analysis, and the final gene set of the clusters was used as the pan gene set. The sequences present in each sample in the cluster in the extracted clustering results were the core gene set, the gene set that was unique to each sample was the special gene set and the pan gene set with the removal of the core gene set and the unique gene set was the dispensable gene set.

#### 2.6.4. Gene Family Analysis

Gene family clustering analysis was performed on MLY102 and the reference strains CFBP2957, CMR15, GMI1000, PSI07, Po82, RS-CIAT078, RS-Rs5, RS-SL2064, RS-T95 and RS-UW251. The protein sequences were subjected to BLAST comparison to remove redundancy, and the gene families were then clustered. After a multiple sequence comparison of clustered gene families using Muscle v3.8.31 [39,40], gene families were analyzed with TreeBeST v1.9.2 using the Neighbor-Joining method for tree-building (parameters: treebest nj -b 1000) [41].

#### 2.6.5. Species Evolution Analysis

A phylogenetic tree was constructed on the basis of the single-nucleotide polymorphism (SNP) matrices of MLY102 and the reference strains CFBP2957, CMR15, GMI1000, PSI07, Po82, RS-CIAT078, RS-Rs5, RS-SL2064, RS-T95, and RS-UW251. The phylogenetic tree was constructed using the Maximum Likelihood algorithm of TreeBeST v1.9.2 (parameters: treebest phyml -b 1000) [41].

## 3. Results and Discussion

### 3.1. Morphological and Biochemical Characteristics of R. solanacearum MLY102

*R. solanacearum* MLY102, which was isolated from morbid tobacco stalks, was pink in color at the center of the colony on TTC medium and was surrounded by a milky white liquid with fluidity (Figure 1A), which was generally consistent with the morphology of a single colony of *R. solanacearum* on TTC medium [42]. When *R. solanacearum* invades the xylem, more extracellular polysaccharides would promote the colonization of *R. solanacearum* and inhibit the conduit water flow, which would ultimately lead to tobacco death [43,44]. Scanning electron microscopy revealed that *R. solanacearum* MLY102 is short and rod-shaped and that the size of the organism is 0.7–0.8 μm × 1.6–1.8 μm (Figure 1B) [5].

Table 1 summarizes the biochemical characteristics of *R. solanacearum* MLY102, as determined via the VITEK 2 Compact automated bacterial identification system using GN cards. The carbon sources that *R. solanacearum* MLY102 was able to utilize included D-glucose (dGLU), cane sugar (SAC) and D-trehalose dihydrate (dTRE) but not D-cellobiose (dCEL), the fermentation of glucose (OFF), D-maltose monohydrate (dMAL), D-mannose (dMNE), paleo sugar (PLE) or D-tagatose (dTAG). SAC and dTRE are considered necessary in *R. solanacearum* infection [45]. *R. solanacearum* may use SAC and dTRE in plants to increase its infestation capacity [46]. It is also worth noting that in addition to carbon sources, *R. solanacearum* infestation led to reductions in ammonium nitrogen in soil and total nitrogen in tobacco [47].

### 3.2. Whole Genome of R. solanacearum MLY102

The genome of *R. solanacearum* strain MLY102 was sequenced via PacBio (8,738,130 reads; 1293 Mb clean data) and DNBSEQ (372,737 reads, with an N50 of 10,408 bp for a total of 3552,404,998 bp) platforms. The genome was assembled, and the assembly resulted in a circular chromosome and a circular plasmid (Figure 2). The size of the genome was 5.72 Mb, which was smaller than that of CQPS-1 (5.89 Mb), FJ1003 (5.90 Mb) and gd-2 (5.93 Mb) isolated from tobacco [1].

The depth of the genome was 43.10× with a G+C content of 67.59%. A total of 5283 protein-coding genes were identified via gene prediction (Figure 3), similar to gd-2 (5074), FJ1003 (5010) and CQPS-1 (5229) [48].

The non-coding RNA (ncRNA) gene characteristics of *R. solanacearum* MLY102 are shown in Table 2. The amount of tRNA was similar to that of gd-2 (59) and CQPS-1 (58) and greater than that of FJ1003 (35) [49]. In addition, 308 tandem repeat fragments, 211 minisatellite DNA sequences, and 33 microsatellite DNA sequences were identified. The different characteristics possessed by strains isolated from tobacco show the complex diversity of *R. solanacearum* [50,51]. This provides additional key evidence at the genetic level for the management of tobacco bacterial wilt disease prevention and control.

### 3.3. Gene Function Annotation of R. solanacearum MLY102

The total number of annotated genes reached 5202 (98.46%), with gene annotation rates of 9.06% (VFDB), 0.51% (ARDB), 2.04% (CAZy), 79.57% (IPR), 38.06% (Swiss-Prot), 73.14% (COG), 0.05% (CARD), 57.08% (GO), 57.9% (KEGG), 97.95% (NR) and 15.65% (T3SS). Among the COG, GO and KEGG terms, the top three terms were amino acid transport and metabolism (401), transcription (389) and general function prediction only (327); cellular process (1759), catalytic activity (1686) and metabolic process (1679); and global and overview maps (1020), amino acid metabolism (284) and carbohydrate metabolism (275) (Figure 4). By screening the VFDB and CARD genes, we identified four hundred and seventy-nine candidate virulence factors and three possible antibiotic resistance genes. The identification of virulence factors and antibiotic resistance genes facilitated the understanding of virulence mechanisms and drug design [51]. The virulence factors mainly include anti-inflammatory proteins and heat shock protein (Hsp) 60, flagellum, and Cya; antibiotic resistance genes are mainly related to antibiotic inactivation and antibiotic efflux and include Vibrio cholerae varG and adeF, which are similar to *R. solanacearum* gd-2 [1]. This provides a key basis for understanding the development of pathogenicity and resistance, thereby supporting the prevention, control and management of tobacco bacterial wilt disease [42].

### 3.4. Comparison of R. solanacearum MLY102 and Reference Strains

#### 3.4.1. Structural Variation and Average Nucleotide Identity (ANI) Analysis of MLY102 and Reference Strains

By comparing MLY102 with the reference strains, structural variations can be used to detect gene positional variations caused by gene recombination and migration. A comparison of the differences between the strains at the nucleotide level (Appendix A) and amino acid level (Appendix A) revealed that MLY102 and the reference strains CFBP2957, CMR15, GMI1000, PSI07, Po82, RS-CIAT078, RS-Rs5, RS-SL2064, RS-T95 and RS-UW251 have homology. ANI was analyzed using fastANI v1.32 (parameters: -c 1024) [37]. The ANI results also revealed similarity between MLY102 and the reference strains (Figure 5). Although the ANI does not rigorously measure the evolutionary relevance of the core genome, as immediate homologs may differ significantly between compared genome pairs, it is able to closely reflect the relevance of DNA–DNA hybridization used to define species in traditional microbiological concepts because it takes into account the mobility of the bacterial gene pool. It also indirectly considers the sharing of functionality between bacteria. Both the MLY102 strain and *R. solanacearum* GMI1000 have ANI values above 99%, indicating a close genetic relationship between them [37]. MLY102 can be identified as phylotype I.

#### 3.4.2. Analysis of Core and Pan Genomes of Eleven Strains

The core and pan genomes were analyzed using CD-HIT v4.6.6 (parameters: -c 0.5 -n 3 -p 1 -g 1 -d 0) [38]. The core genome is needed to maintain the basic biological characteristics of all strains [52]. A sequence comparison of eleven strains of *R. solanacearum* revealed 4378 homologous gene groups (HGGs), 2273 of which were shared by eleven genomes, constituting the core genome of *R. solanacearum* (Figure 6B) [5]. The number of genes specific to each strain varied widely, and these differences support the high complexity and genomic diversity of *R. solanacearum* [51]. In addition, a heatmap of dispensable genes was drawn on the basis of the distribution of dispensable genes in different samples (Figure 6A). The results revealed that CFBP2957 was independent of other branches and presented unique properties; MLY102 and GMI1000 clustered into a cluster and clustered into a branch with CMR15, which expressed a high degree of similarity. PSI07, RS-SL2064 and RS-T95 formed a branch. Another important branch was formed by Po82, RS-CIAT078, RS-Rs5 and RS-UW251. The pan gene set Venn diagram and statistical graph of the number of gene family homologs revealed that, compared with the reference strain, MLY102 presented the second highest number of special genes (493) and the greatest number of species-specific gene family homologs (13) (Figure 6C). Dispensable genes and special genes play important roles in the genetic diversity of species and in adverse environments [5]. These genes may provide important clues for understanding the species evolution and biodiversity of *R. solanacearum*.

#### 3.4.3. Evolutionary Relationship Between MLY102 and Reference Strains

We constructed phylogenetic trees using the Maximum Likelihood method (parameters: treebest phyml -b 1000) [41]. The results of core pan and gene family analyses revealed that RS-SL2064 clustered with RS-T95; Po82 clustered with RS-CIAT078 (Figure 7). This is consistent with the analysis results of ANI and the dispensable gene heatmap. MLY102 had the maximum ANI value (99.01%) with only GMI1000 among all the reference strains and was clustered into the same branch in the dispensable gene heatmap, whereas, in sharp contrast, the two strains were classified into different main branches in the phylogenetic tree.

## 4. Conclusions

The identification and genomic analysis of the R. solanacearum strain MLY102 provided significant data on the genetic evolutionary relationships of R. solanacearum. MLY102 is able to utilize specific carbon sources, which provide the basis for its ability to infest plants. Comprehensive genome sequencing demonstrated the unique genetic characteristics of R. solanacearum, including a single megaplasmid (and, in some cases, a single miniplasmid). In addition, comparative genetic analyses of eleven different strains of R. solanacearum provided important clues to the diversity and phylogenetic relationships of R. solanacearum. Genomic information can be applied in the field for early detection, which is beneficial for disease detection and management. In addition, comparative genomic studies of different strains of R. solanacearum can be expanded in future studies, which will help reveal differences in the adaptability of R. solanacearum in different ecosystems and provide a scientific basis for regional management of the disease.

## Figures and Tables

**Figure 1 genes-15-01473-f001:**
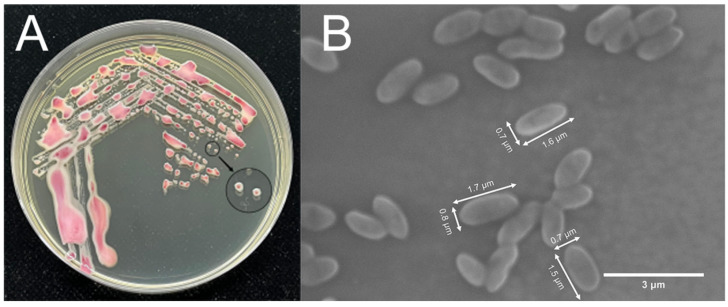
Colony appearance (**A**) and cellular morphology (**B**) of *R. solanacearum* MLY91.

**Figure 2 genes-15-01473-f002:**
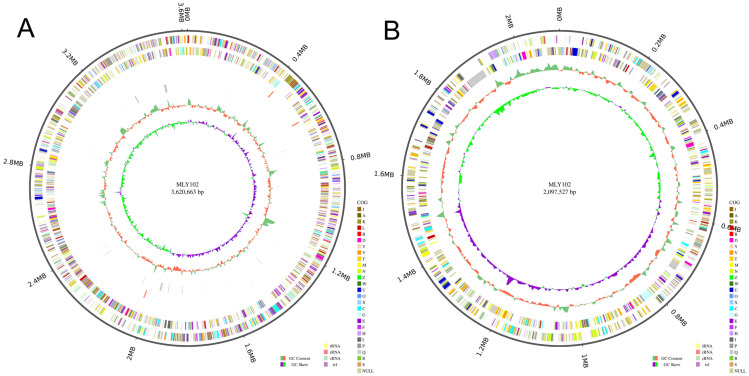
Circle graphs of the *R. solanacearum* MLY102 chromosome (**A**) and giant plasmid (**B**).

**Figure 3 genes-15-01473-f003:**
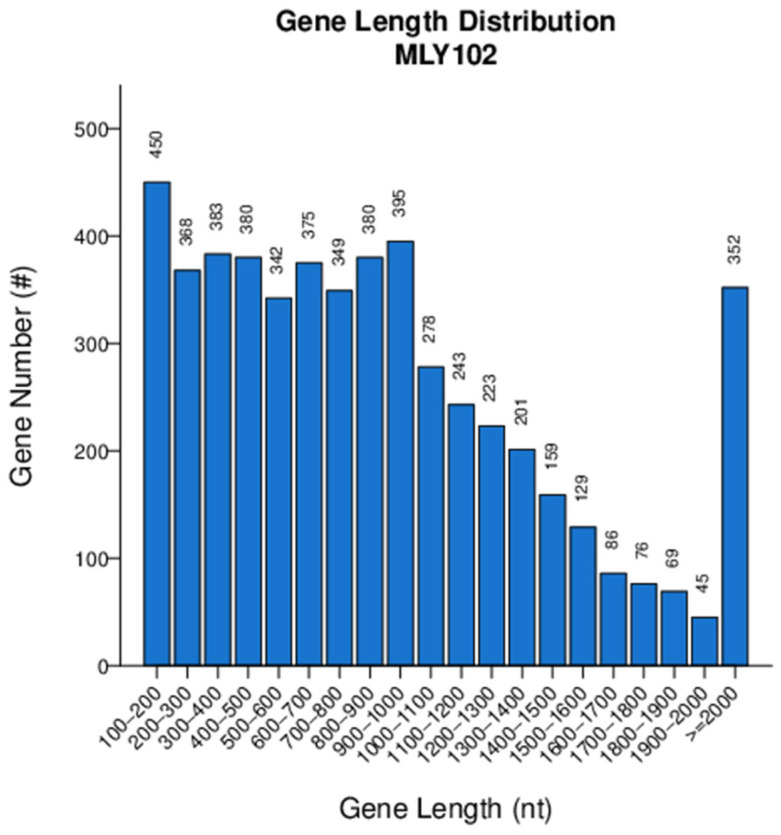
Distribution of gene lengths of *R. solanacearum* MLY102.

**Figure 4 genes-15-01473-f004:**
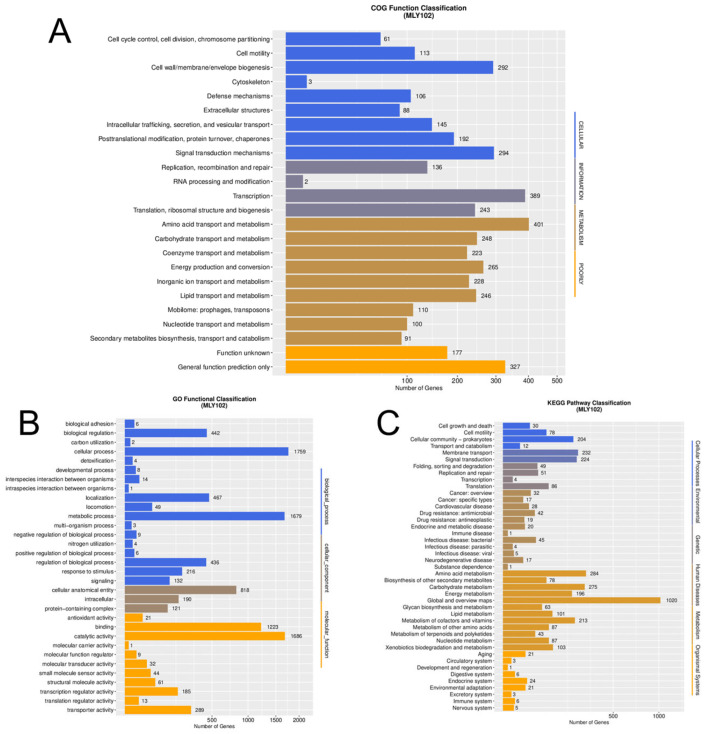
Functional classification of genes in *R. solanacearum* MLY102. (**A**): COG function classification; (**B**): GO function classification; (**C**): KEGG function classification.

**Figure 5 genes-15-01473-f005:**
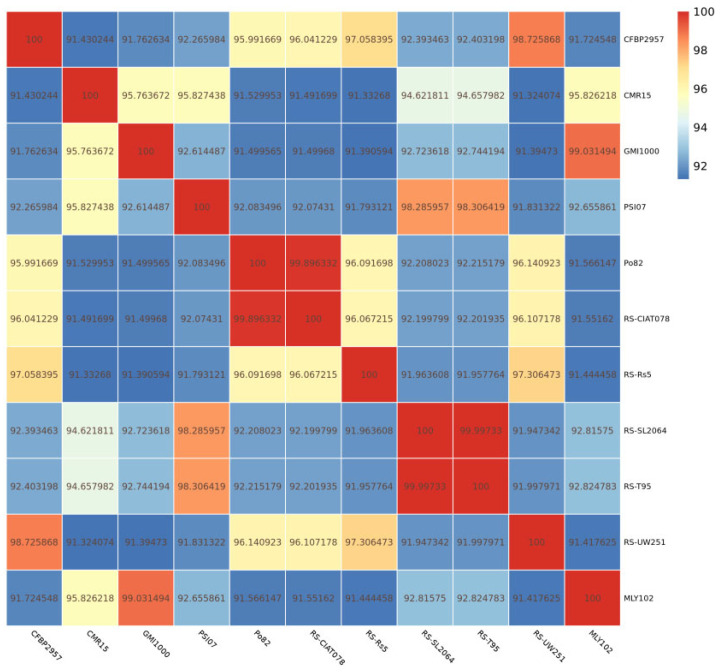
Heatmap of average nucleotide identity (ANI) between eleven strains of *R. solanacearum*.

**Figure 6 genes-15-01473-f006:**
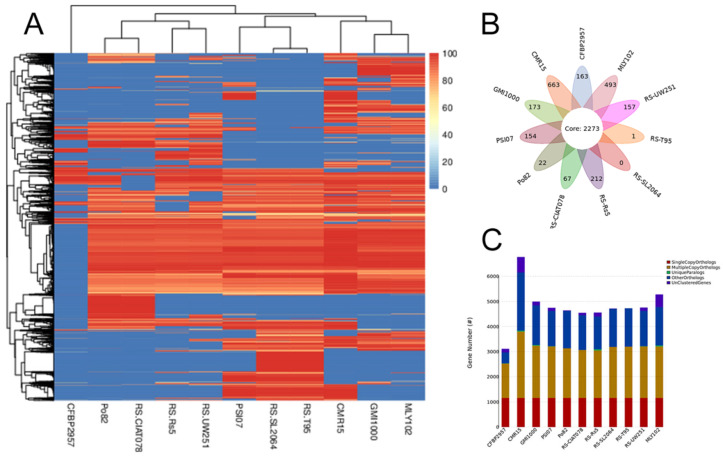
Heatmap of dispensable genes (**A**), Venn diagram of the pan gene set (**B**) and statistics of the number of homologous genes in the gene families (**C**) of *R. solanacearum* MLY102 and the reference strains.

**Figure 7 genes-15-01473-f007:**
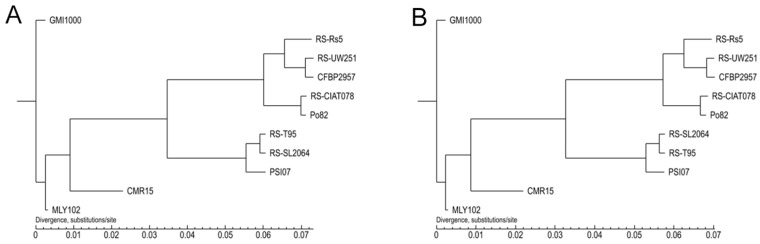
Phylogenetic trees of *R. solanacearum* MLY102 and reference strains based on core pan (**A**) and gene family (**B**) results.

**Table 1 genes-15-01473-t001:** Biochemical characteristics of *R. solanacearum* MLY102.

Well No.	Tests	Results	Well No.	Tests	Results	Well No.	Tests	Results
2	APPA	-	21	BXYL	-	41	AGLU	-
3	AOD	-	22	BAIap	-	42	SUCT	+
4	PyrA	-	23	ProA	+	43	NAGA	-
5	IARL	-	26	LIP	-	44	AGAL	-
7	dCEL	-	27	PLE	-	45	PHOS	-
9	BGAL	-	29	TyrA	+	46	GlyA	-
10	H2S	-	31	URE	+	47	ODC	-
11	BNAG	-	32	dSOR	+	48	LDC	-
12	AGLTp	-	32	dMNE	-	53	IHISa	-
13	dGLU	+	33	SAC	+	56	CMT	-
14	GGT	-	34	dTAG	-	57	BGUR	-
15	OFF	-	35	dTRE	+	58	O129R	-
17	BGLU	-	36	CIT	+	59	GGAA	-
18	dMAL	-	37	MNT	-	61	IMLTa	-
19	dMAN	-	39	5KG	-	62	ELLM	-
20	dMNE	-	40	ILATk	-	64	ILATa	-
			GN Card			Incubation time: 7.25 h		

“+” indicates that the bacterium is capable of producing a certain reaction or possessing a certain characteristic under certain conditions; “-” indicates that the bacterium is unable to produce a reaction or does not possess a characteristic under certain conditions.

**Table 2 genes-15-01473-t002:** Genomic characterization of the non-coding RNA (ncRNA) *R. solanacearum* MLY102.

Type	Number	Average Length (bp)	Size (bp)
tRNA	61	79.88	4873
5 s_rRNA	4	112	448
16 s_rRNA	4	1524	6096
23 s_rRNA	4	2877	11,508
sRNA	22	75.27	1656

## Data Availability

The MLY102 genome sequence was deposited in the DDBJ/ENA/GenBank database (BioProject/BioSample PRJNA1177238/SAMN44414988).

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
