# Peer review of "Whole-Genome Sequence and Characterization of Ralstonia solanacearum MLY102 Isolated from Infected Tobacco Stalks"

_genes, 2024, doi:10.3390/genes15111473_

Round 1
Reviewer 1 Report
Comments and Suggestions for Authors
Dear Authors,
The manuscript titled “Whole-genome sequence and characterization of Ralstonia solanacearum MLY102 isolated from tobacco morbidly infected tobacco stalks” aimed to isolate and characterize the Ralstonia solanacearum strain MLY102, responsible for bacterial wilt disease in tobacco. Isolated from an infected stalk, MLY102 showed a unique pinkish-red center with a milky-white perimeter on TTC medium. It can utilize D-glucose, cane sugar, and D-trehalose dihydrate as carbon sources.
Genome sequencing revealed a 5.72 Mb genome with 5,283 protein-coding genes, including unique genetic elements, distinguishing it from closely related strains GMI1000 and CMR15. MLY102’s distinctive genomic features provide valuable insights for advancing bacterial wilt disease prevention and control strategies.
In reviewing this manuscript, several key areas emerge where revisions could enhance clarity, coherence, and impact:
1. Abstract and Introduction:
- The abstract contains details that may obscure its primary findings and objectives. Simplifying complex information and focusing on major findings would improve readability.
- In the introduction, the study's objectives could be stated more explicitly and directly. Additional context about the significance of Ralstonia solanacearum in tobacco farming could reinforce the study's importance.
2. Materials and Methods:
- While the methods section is thorough, some areas contain excessive procedural details that could be streamlined. Breaking complex processes into clear subsections or flowcharts may aid readers.
- Ensure that all methods use standardized scientific terms and are concise. Overly specific information, such as equipment brands and catalog numbers, might be unnecessary unless directly impacting results.
3. Results Section:
- Figures and tables appear useful, yet some could benefit from clearer legends or descriptions. Aligning figures more closely with the text where they are discussed can help readers better follow results.
- It is unclear if all data received statistical validation. Highlighting statistical significance and analysis methods where applicable would support the rigor of the findings.
4. Discussion and Interpretation:
- The discussion could deepen its comparative analysis by linking findings to broader implications for bacterial wilt management in tobacco. Citing recent studies could strengthen this context.
- The conclusions summarize findings well but could extend on potential applications in agriculture or disease management. Outlining future research directions could also add depth.
5. Language and Style:
- Certain terms are complex and may need further clarification or simplification for broader accessibility. Reducing jargon and using straightforward language, where possible, would enhance readability.
- Enhancing transitions between sections could improve cohesion. Each section should smoothly lead to the next, ensuring logical flow from one concept to another.
6. Figures and Tables:
- Ensuring high-resolution images and consistent, clear labeling for figures and tables will make the manuscript visually accessible.
- More descriptive figure legends could help readers understand each figure independently of the text.
7. References and Citations:
- Including recent literature, particularly from the past two years, would add current perspectives. Consistent citation formatting is essential to maintain professional standards.
Author Response
Comments 1: The abstract contains details that may obscure its primary findings and objectives.Simplifying complex information and focusing on major findings would improve readability.
Response 1: Thank you for your comments. Therefore, we have added “which causes huge losses to crop economies worldwide” to explain the background of the study and to emphasise the aim of the study. We have added “The genomic properties of MLY102 were explored by performing biochemical characterisation, genome sequencing, compositional analysis, functional annotation and comparative genomic analysis” to describe the background of the study and highlight the research objectives. We have deleted the phrases “ in a plot with a high incidence of bacterial wilt disease for several years” and “308 tandem repeat fragments, 211 minisatellite DNA sequences, and 33 microsatellite DNA sequences” to simplify the complex information.
Comments 2: In the introduction,the study's objectives could be stated more explicitly and directly.Additional context about the significance of Ralstonia solanacearum in tobacco farming could reinforce the study's importance.
Response 2: We agree with this comment. In the first paragraph of the introduction, we have added “The infestation of tobacco with R. solanacearum causes significant economic losses and yield reductions, contributing to 10-30% of production losses globally”; In the last paragraph of the introduction, the following text was added: “which will provide key data for the development of new control strategies. This is not only crucial for the sustainable development of the tobacco industry, but also has broad implications for disease prevention and control in other crops” to reinforce the importance of research.
Comments 3: While the methods section is thorough,some areas contain excessive procedural details that could be streamlined.Breaking complex processes into clear subsections or flowcharts may aid readers.
Response 3: Agree. We have streamlined some complicated procedural details, including modifying “Following the manufacturer’s instructions...After incubation, the VITEK® 2 Compact system was used to analyze the biochemical reaction patterns” to “and the biochemical reaction pattern was analyzed using the VITEK® 2 Compact System” in Section 2.2.2; modifying “Genomic DNA was extracted from freshly grown MLY02 via the MGIEasy Microbiome DNA Extraction Kit (MGI Tech, China) according to the manufacturer's instructions” to “DNA from MLY102 was extracted using the MGIEasy Microbiome DNA Extraction Kit” in Section 2.3; and in Section 2.6.1, streamlining the operation steps by modifying “and then, after the sequences...and the consistent value of this hit was the average of the consistent values of the two comparisons” to “Protein comparison was a BLASTp comparison of the target bacterial protein set to the reference bacterial protein set”. Simplified description of the tree-building analysis method in section 2.6.4: “After a multiple sequence comparison of clustered gene families using Muscle v3.8.31...”. In section 2.6.5, removed the unimportant information “For each strain, all SNPs were linked in the same order to obtain sequences of the same length in fasta format as input files”. In addition, Supplementary Figure S1 was added to the last paragraph of the introduction to make the complex process clear. All changes are highlighted in the Marked up manuscript file.
Comments 4: Ensure that all methods use standardized scientific terms and are concise.Overly specific information,such as equipment brands and catalog numbers,might be unnecessary unless directly impacting results.
Response 4: Agree. The overly specific information has been removed from Comments 3. We have also changed the terminology from “HiFi libraries” to “high-fidelity reads” in Section 2.3; In section 2.6.4, “NJ method” is changed to “Neighbor-Joining method”.
Comments 5: Figures and tables appear useful,yet some could benefit from clearer legends or descriptions.Aligning figures more closely with the text where they are discussed can help readers better follow results.
Response 5: We agree with this comment. We have adjusted the position between the text and the graphs in Section 3.1, and increased the resolution of Figures 2 and 4 to make them clearer.
Comments 6: It is unclear if all data received statistical validation.Highlighting statistical significance and analysis methods where applicable would support the rigor of the findings.
Response 6: Agree. We have added analysis methods in Materials and Methods and in Results and Discussion, for example, “ANI was analysed using fastANI v1.32 (parameters: -c 1024) [37]” was added in Section 3.4.1; Section 3.4.2 added “The core and pan genomes were analysed using CD-HIT v4.6.6 (parameters: -c 0.5 -n 3 -p 1 -g 1 -d 0) [38]”; Section 3.4.3 added “We constructed phylogenetic trees using the Maximum Likelihood method (parameters: treebest phyml -b 1000) [41]”.
Comments 7:The discussion could deepen its comparative analysis by linking findings to broader implications for bacterial wilt management in tobacco.Citing recent studies could strengthen this context.
Response 7: We agree with this comment. To this end we have added “When R. solanacearum invaded the xylem, more extracellular polysaccharides would promote the colonisation of R. solanacearum and inhibit the conduit water flow, which would ultimately lead to tobacco death [43,44]” and “It was also worth noting that in addition to carbon sources, R. solanacearum infestation leaded to reductions in ammonium nitrogen in soil and total nitrogen in tobacco [47]”;
In section 3.2, “The different characteristics possessed by strains isolated from tobacco show the complex diversity of R. solanacearum [50,51]. This provides additional key evidence at the genetic level for the management of tobacco bacterial wilt disease prevention and control” was added to deepen the discussion on the relationship between bacterial wilt disease and tobacco wilt.
Comments 8: The conclusions summarize findings well but could extend on potential applications in agriculture or disease management.Outlining future research directions could also add depth.
Response 8: We agree with this comment. Therefore, we added “Genomic information can be applied in the field for early detection, which is beneficial for disease detection and management. In addition, comparative genomic studies of different strains of R. solanacearum can be expanded in future studies, which will help reveal differences in the adaptability of R. solanacearum in different ecosystems and provide a scientific basis for regional management of the disease” to the conclusion.
Comments 9: Certain terms are complex and may need further clarification or simplification for broader accessibility.Reducing jargon and using straightforward language,where possible,would enhance readability.
Response 9: Thank you for your advice. We use MDPI's language editing to make changes. In addition, some technical terms are explained. For example, "HiFi libraries" in section 2.3 is changed to "high-fidelity reads"; In Section 2.6.4, "NJ method" is changed to "Neighbor-Joining method".
Comments 10: Enhancing transitions between sections could improve cohesion.Each section should smoothly lead to the next,ensuring logical flow from one concept to another.
Response 10: Agree. To better enable the reader to understand the process, we have added Figure S1.
Comments 11: Ensuring high-resolution images and consistent,clear labeling for figures and tables will make the manuscript visually accessible.
Response 11: Agree. We have increased the resolution of Figures 2 and 4 to make them clearer.
Comments 12: More descriptive figure legends could help readers understand each figure independently of the text.
Response 12: Agree. We have modified the titles of Figures 4 and 5 to “Figure 4. Functional classification of genes in R. solanacearum MLY102. A: COG function classification; B: GO function classification; C: KEGG function classification” and “Figure 5. Heatmap of average nucleotide identity (ANI) between eleven strains of R. solanacearum” to make them easier to understand.
Comments 13: Including recent literature,particularly from the past two years,would add current perspectives.Consistent citation formatting is essential to maintain professional standards.
Response 13: We agree with this comment. We added nine papers from the last two years and standardised the citation format.

Reviewer 2 Report
Comments and Suggestions for Authors
The purpose of the study was to evaluate the whole-genome sequence and characterization of Ralstonia solanacearum MLY102 from tobacco infected with tobacco stalks. The Authors unveiled that MLY102 may utilize D-glucose (dGLU), cane sugar (SAC) and D-trehalose dihydrate (dTRE). Genome sequencing through the DNBSEQ and PacBio platforms revealed a genome size of 5.72 Mb with a 67.59% G+C content. The genome consists of a circular chromosome and a circular giant plasmid with 5,283 protein-coding genes, 308 tandem repeat fragments, 211 minisatellite DNA sequences, and 33 microsatellite DNA sequences. In addition, a comparison of the genomes revealed that MLY102 is closely related to GMI1000 and CMR15 but has 498 special genes and 13 homologous genes in the species-specific gene family, indicating a high degree of genomic uniqueness. In my opinion, the paper presents important research findings and conclusions in the frame of the topic. In order to increase understanding the paper by readers and to improve its layout and scientific level, I suggest the following minor revisions:
- English style and grammar should be checked out and improved by the native speaker, specialist in the research discipline,
- Statistical analyses of the results should be described in more detail,
- Fig. 2 (Circle graphs of the Ralstonia solanacearum MLY102 chromosome (A) and giant plasmid (B)) is almost unreadable because of low graphical resolution. Hence, I recommend increasing the resolution to present these results in a more visible way,
- Fig. 4 (Functional classification of genes in R. solanacearum MLY102. A: COG; B: GO; C: KEEG) – the same situation as described above.
Author Response
Comments 1: English style and grammar should be checked out and improved by the native speaker,specialist in the research discipline,
Response 1: Agree. We have had the article checked and improved by MDPI's language editor.
Comments 2: Statistical analyses of the results should be described in more detail,
Response 2: Agree. We have added analysis methods in Materials and Methods and in Results and Discussion, for example, “ANI was analysed using fastANI v1.32 (parameters: -c 1024) [37]” was added in Section 3.4.1; Section 3.4.2 added “The core and pan genomes were analysed using CD-HIT v4.6.6 (parameters: -c 0.5 -n 3 -p 1 -g 1 -d 0) [38]”; Section 3.4.3 added “We constructed phylogenetic trees using the Maximum Likelihood method (parameters: treebest phyml -b 1000) [41]”. In addition, we have added the section of the Results and Discussion with tobacco bacterial wilt disease, highlighted in the Marked Up Manuscript file.
Comments 3: Fig.2(Circle graphs of the Ralstonia solanacearum MLY102 chromosome(A)and giant plasmid(B))is almost unreadable because of low graphical resolution.Hence,I recommend increasing the resolution to present these results in a more visible way,
Response 3: Agree. We have replaced the image with a higher resolution.
Comments 4: Fig.4(Functional classification of genes in R.solanacearum MLY102.A:COG;B:GO;C:KEEG)–the same situation as described above.
Response 4: Agree. We have replaced the image with a higher resolution and modified the titles of Figures 4 and 5 to “Figure 4. Functional classification of genes in R. solanacearum MLY102. A: COG function classification; B: GO function classification; C: KEGG function classification” and “Figure 5. Heatmap of average nucleotide identity (ANI) between eleven strains of R. solanacearum” to make them easier to understand.

Reviewer 3 Report
Comments and Suggestions for Authors
Dear Authors,
Reviewer comments genes-3305771
The manuscript entitled „Whole-genome sequence and characterization of Ralstonia solanacearum MLY102 isolated from infected tobacco stalks“ represents a useful study aimed at an isolation, identfication and sequencing of Ralstonia solanacearum MLY102 from tobacco. The presented genomic data on R. solanacearum are surely worth publishing. I have only a few minor comments on the present manuscript which are given below:
In the manuscript title, remove the words „tobacco morbidly“ and modify the title as follows: „Whole-genome sequence and characterization of Ralstonia solanacearum MLY102 isolated from infected tobacco stalks“.
Materials and methods, line 117: Modify the word „less“ to „lower“ in the statement: „Subreads lower than 1000 bp in length were filtered…“
Figure 4C legend, line 238: Correct the term „KEGG“ (not „KEEG“) for an online database used for R. solanacearum MLY102 functional annotation.
The section entitled „Results and Discussion“ is mostly just presentation of Results. I think that the discussion section has to be newly written and more literature references have to be included.
Final recommendation: Accept after a minor revision.
Author Response
Comments 1: In the manuscript title, remove the words „tobacco morbidly“ and modify the title as follows: „Whole-genome sequence and characterization of Ralstonia solanacearum MLY102 isolated from infected tobacco stalks“
Response 1: Agree. We have changed the title to “Whole-genome sequence and characterization of Ralstonia solanacearum MLY102 isolated from infected tobacco stalks”.
Comments 2: Materials and methods, line 117: Modify the word „less“ to „lower“ in the statement: „Subreads lower than 1000 bp in length were filtered…“
Response 2: We agree with this comment. We have changed it to “Subreads lower than 1000 bp in length were filtered…”.
Comments 3: Figure 4C legend, line 238: Correct the term „KEGG“ (not „KEEG“) for an online database used for R. solanacearum MLY102 functional annotation.
Response 3: Agree. We have corrected and modified the title of the Figure as “Figure 4. Functional classification of genes in R. solanacearum MLY102. A: COG function classification; B: GO function classification; C: KEGG function classification” makes it easier for readers to understand.
Comments 4: The section entitled „Results and Discussion“ is mostly just presentation of Results. I think that the discussion section has to be newly written and more literature references have to be included.
Response 4: Agree. We discussed the results in more depth and included 9 additional references from the past two years. For example, we added “which was basically consistent with the morphology of single colony of R. solanacearum on TTC medium [42]”, “When R. solanacearum invaded the xylem, more extracellular polysaccharides would promote the colonisation of R. solanacearum and inhibit the conduit water flow, which would ultimately lead to tobacco death [43,44]” and “It was also worth noting that in addition to carbon sources, R. solanacearum infestation leaded to reductions in ammonium nitrogen in soil and total nitrogen in tobacco [47]” in Section 3.1 to deepen the discussion of the relationship between solanachum and tobacco bacterial wilt. In addition, add “ The size of the genome was 5.72 Mb, which was smaller than that of CQPS-1 (5.89Mb), FJ1003 (5.90Mb) and gd-2 (5.93 Mb) isolated from tobacco [1]”, “ similar to gd-2 (5074), FJ1003 (5010) and CQPS-1 (5229) [48]”, “The amount of tRNA was similar to that of gd-2 (59) and CQPS-1 (58) and greater than that of FJ1003 (35) [49]” and “The different characteristics possessed by strains isolated from tobacco show the complex diversity of R. solanacearum [50,51]. This provides additional key evidence at the genetic level for the management of tobacco bacterial wilt disease prevention and control” in Section 3.2 to illustrate the complex diversity of solanacearum. In Section 3.3, we added “The identification of virulence factors and antibiotic resistance genes facilitated the understanding of virulence mechanisms and drug design [51]” and “This provides a key basis for understanding the development of pathogenicity and resistance, thereby supporting the prevention, control and management of tobacco bacterial wilt disease [42]” to indicate its pathogenicity and resistance. All modifications are highlighted in the Marked up manuscript document.
